# Impact of Hydroxychloroquine Treatment of COVID-19 on Cardiac Conduction: The Beat Goes On

**Marc Thomas Zughaib \*, Robby Singh, Marcel Letourneau**  **and Marcel Elias Zughaib**

Ascension Providence Hospital Heart Institute, Michigan State University, East Lansing, MI 48075, USA;
robby9697@gmail.com (R.S.); mletourneau51@gmail.com (M.L.); mzughaib@comcast.net (M.E.Z.)
\* Correspondence: marc.zughaib@ascension.org; Tel.: +1-248-953-4909

**Abstract:** Objectives: Our study aimed to investigate the frequency of malignant cardiac arrhythmias in hospitalized patients receiving hydroxychloroquine alone and those receiving a combination of hydroxychloroquine with azithromycin, as well as the quantitative extent of QT prolongation within Tisdale Risk Score (TRS) categories. Background: There have been over 33 million cases of SARS-CoV-2 (COVID-19) resulting in over 600,000 deaths in the United States. As the current COVID-19 pandemic continues, numerous medications have been administered to attempt to treat patients afflicted by the disease. While hydroxychloroquine has been in use for decades for rheumatologic and infectious disease processes, it does have potential cardiotoxicity related to drug-induced QT prolongation. Drug-induced QT prolongation has an increased risk of arrhythmogenicity, potentially progressing into torsades de pointes (TdP) and increased patient mortality. The relationship between QT prolongation and TdP is complex and inexact, but there remains optimism regarding the use of these medications in the treatment of COVID-19 despite limited data on their true efficacy. Methods: We retrospectively identified 75 patients who were admitted with COVID-19 and underwent treatment with hydroxychloroquine for 5 days. The hydroxychloroquine protocol was defined as an initial dose of 400 mg BID for the first day, followed by 400 mg daily for the next 4 days. Baseline demographics, medications, medical histories, lab values, ECG QT intervals, and Tisdale Risk Categories were collected for all patients. Results: Seventy-four (98.7%) patients completed the full course of hydroxychloroquine. There were 41 males (54.7%) and 34 females (45.3%). Average length of stay was 8.9 days (95% CI: 7.5, 10.2). One patient who could not complete the course due to inability to swallow medication tablets. There were no reports of new arrythmias or incidence of torsades de pointes during the study. Seventy-two patients (96%) were taking at least 2 QT prolonging medications. The average corrected QT intervals were as follows: day 1 of admission was 421.62 milliseconds ($n = 66$, 95% CI: 412.19, 431.05), day 2 was 431.50 ms ($n = 30$, 95% CI: 416.34, 446.66), day 3 was 433.48 ms ($n = 23$, 95% CI: 413.34, 453.61), day 4 was 427.59 ms ($n = 17$, 95% CI: 400.83, 454.35), and day 5 was 444.28 ms ($n = 18$, 95% CI: 428.43, 460.12). The corrected QT interval prolonged by 22.66 ms from day 1 to day 5 ($p = 0.03$) in the overall population. Conclusion: There were no patients who experienced arrhythmogenicity or Torsades de Pointes despite a statistically significant increase in QTc intervals after patients received the 5-day course of hydroxychloroquine for treatment of COVID-19.

**Keywords:** COVID-19; hydroxychloroquine; QTc

## 1. Introduction

There have been over 43 million cases of SARS-CoV-2 (COVID-19) resulting in over 690,000 deaths in the United States [1]. As the current COVID-19 pandemic continues, treatment regimens have been administered to attempt to treat patients afflicted by the disease. There are no medications that have received FDA approval to prevent or treat COVID-19 [2]. Several therapeutic regimens in the United States have been used to treat these patients including chloroquine, hydroxychloroquine, remdesivir, azithromycin, convalescent plasma, and prednisone [3]. There is limited data from randomized controlled

trials (RCTs) to inform clinicians on the benefits, dosing, duration, and adverse effects of these medications for treatment or prophylaxis of COVID-19 infection [4–7].

Among these medications, hydroxychloroquine has been studied as a potential treatment of the virus [8–12]. The therapeutic benefit of hydroxychloroquine as a treatment for COVID-19 has been investigated by several studies, with mixed results [12–16]. While hydroxychloroquine has been in use for decades for malaria and rheumatologic conditions, it can display potential cardiotoxicity with patients experiencing drug-induced QT prolongation. Drug-induced QT prolongation has an increased risk of arrhythmogenicity, potentially progressing into torsades de pointes (TdP) and increasing patient mortality. The relationship between QT prolongation and TdP is complex. Tisdale et al. derived and validated a QT prolongation risk score comprised of baseline demographic, medical history, and lab data. This score categorizes patients into low, moderate, or high risk of drug-associated QT prolongation among cardiac-care-unit-hospitalized patients [17].

There remains optimism regarding the use of these medications in the treatment of COVID-19 despite limited data on their true efficacy. Our objective is to provide some insight into the safety of the administration of these medications with regard to cardiotoxicity. Our study aims to investigate the frequency of malignant cardiac arrhythmias in hospitalized patients receiving hydroxychloroquine alone and those receiving a combination of hydroxychloroquine with azithromycin, as well as the quantitative extent of QT prolongation within Tisdale Risk Score (TRS) categories.

## 2. Methods

We retrospectively identified patients who were admitted with COVID-19 from March–April 2020 at our facility. Initially, 531 patients were identified who met inclusion criteria, and 75 patients were randomly selected for our study. Inclusion criteria included patients older than 18 years old, were admitted to the hospital with a diagnosis of COVID-19 infection, and underwent treatment with the hydroxychloroquine protocol for 5 days. The hydroxychloroquine protocol was defined as an initial dose of 400 mg BID for the first day, followed by 400 mg daily for the next 4 days. Baseline demographics, medications, medical histories, lab values, and ECG QT intervals were collected for all patients included in this study. QT intervals were calculated by hand on every ECG and compared to the QT interval that was calculated by the computer to confirm exact measurements. Tisdale Risk scores were subsequently calculated for each patient based on the pre-defined and validated criteria. Statistical analyses were performed with Microsoft Excel. Descriptive statistics were performed on the entire study population, as well as within each Tisdale Risk Score group. Continuous data were analyzed using 2 tailed t-tests.

## 3. Results

There were 75 patients who met full inclusion criteria. The average age of the patient population was 67.2 years old (95% CI: 63.9, 70.5) (Table 1). There were 41 males (54.7%) and 34 females (45.3%). Our study included 26 (24.7%) Caucasians, 45 (60%) African Americans, and 4 (5.3%) patients with other/unknown ethnicities. Average BMI was 32.7 (95% CI: 30.7, 34.8). Average length of stay was 8.9 days (95% CI: 7.5, 10.2). There were 63 (84%) patients with hypertension, 38 (50.7%) with diabetes, and 12 (16%) patients were admitted to the ICU.

Out of the 75 patients, 74 (98.7%) completed the full course of hydroxychloroquine (Table 2). There was 1 patient who could not complete the course because they were unable to swallow medication tablets. There were no reports of new arrythmias or incidence of torsades de pointes during the study. There were 72 patients (96%) that were taking at least 2 QT prolonging medications. In total, 68 patients were receiving azithromycin, and 4 other patients were either receiving amiodarone, ondansetron, doxycycline, or paliperidone as the second QT prolonging medications. Using the Tisdale Risk Scoring for drug associated QT prolongation there were 9 patients (12%) that were classified as low risk, 41 patients (55%) as moderate risk, and 25 patients (33%) as high risk for QTc prolongation.

**Table 1.** Baseline characteristics for entire patient population. *n* = 75.

|  | **Mean** | **95% CI** | | **%** |
|---|---|---|---|---|
| Age | 67.18 | 63.90 | 70.46 | |
| Gender (*n*) | | | | |
| - male | 41 | | | 54.7 |
| - female | 34 | | | 45.3 |
| Ethnicity (*n*) | | | | |
| - Caucasian | 26 | | | 34.7 |
| - African American | 45 | | | 60.0 |
| - Other/unknown | 4 | | | 5.3 |
| BMI | 32.72 | 30.66 | 34.77 | |
| Length of Stay | 8.86 | 7.52 | 10.20 | |
| HTN (*n*) | 63 | | | 84.0 |
| DM (*n*) | 38 | | | 50.7 |
| Ventilated (*n*) | 10 | | | 13.3 |
| ICU (*n*) | 12 | | | 16.0 |

**Table 2.** Tisdale Risk Score characteristics of entire patient population. High Risk: ≥11 points, Moderate risk: 7–10 points, Low risk: <6. Maximum risk score is 21 points with variables weighted from 1–3 points.

|  | *n* | % | Points |
|---|---|---|---|
| Completed Course | 74 | 98.67 | 3 |
| Age over 68 | 36 | 48.00 | 1 |
| Female Gender | 34 | 45.33 | 1 |
| Prolong QTc on Admission | 24 | 32.00 | 2 |
| 2 QT prolonging meds | 72 | 96.00 | 3 |
| Acute MI | 0 | 0 | 2 |
| Loop Diuretic? | 23 | 30.67 | 1 |
| >2 SIRS criteria | 49 | 65.33 | 3 |
| Heart failure? | 3 | 4.00 | 3 |
| Serum K < 3.5 mEq/L | 11 | 14.67 | 2 |
| Tisdale Risk Category | | | |
| - low | 9 | 12.00 | |
| - moderate | 41 | 54.67 | |
| - high | 25 | 33.33 | |

The average corrected QT intervals were as follows: day 1 of admission was 421.62 milliseconds (*n* = 66, 95% CI: 412.19, 431.05), day 2 was 431.50 ms (*n* = 30, 95% CI: 416.34, 446.66), day 3 was 433.48 ms (*n* = 23, 95% CI: 413.34, 453.61), day 4 was 427.59 ms (*n* = 17, 95% CI: 400.83, 454.35), and day 5 was 444.28 ms (*n* = 18, 95% CI: 428.43, 460.12) (Figure 1). The corrected QT interval prolonged by 22.66 ms from day 1 to day 5 (*p* = 0.03) in the overall population.

The average corrected QT intervals for patients within the Tisdale High Risk group were as follows: day 1 of admission was 429.68 milliseconds (*n* = 25, 95% CI: 413.43, 445.93), day 2 was 433.87 ms (*n* = 15, 95% CI: 416.88, 450.86), day 3 was 451.14 ms (*n* = 7, 95% CI: 402.15, 500.14), day 4 was 440.22 ms (*n* = 9, 95% CI: 390.69, 489.75), and day 5 was 453.33 ms (*n* = 6, 95% CI: 426.29, 480.37) (Figure 2). The corrected QT interval prolonged by 23.65 ms from day 1 to day 5 (*p* = 0.19) within this risk group.

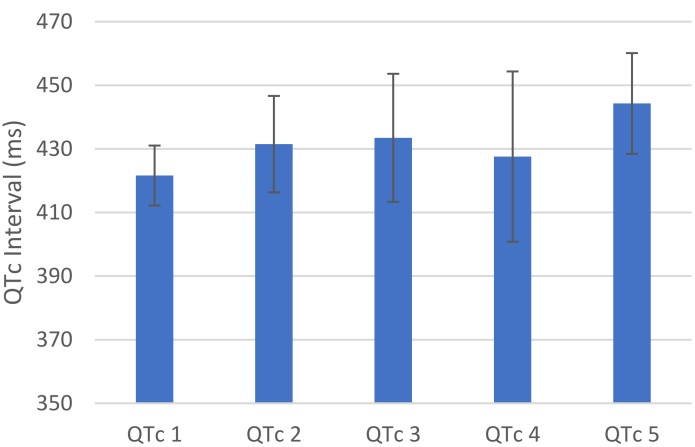

**Figure 1.** Average QTc for entire population. QTc reported in ms, *p* = 0.03 from QTc 1 to QTc 5. Error bars represent 95% confidence interval.

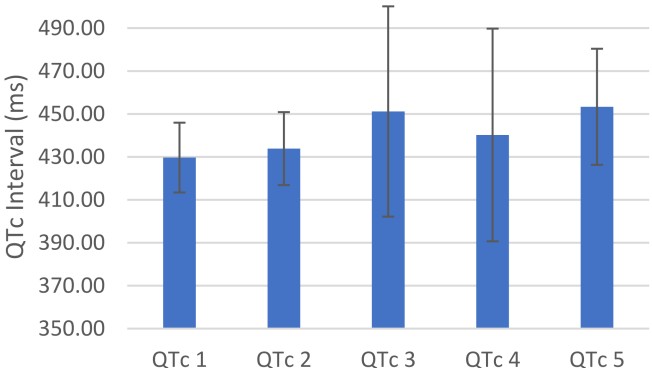

**Figure 2.** Average QTc for patients with High Risk Tisdale Score. QTc reported in ms, *n* = 25, *p* = 0.19 from QTc 1 to QTc 5. Error bars represent 95% Confidence Intervals.

The average corrected QT intervals for patients within the Tisdale Moderate Risk group were as follows: day 1 of admission was 424.61 milliseconds (*n* = 36, 95% CI: 412.10, 437.13), day 2 was 443.25 ms (*n* = 12, 95% CI: 416.74, 469.76), day 3 was 430.83 ms (*n* = 12, 95% CI: 410.58, 451.09), day 4 was 422.20 ms (*n* = 5, 95% CI: 411.51, 432.89), and day 5 was 445.40 ms (*n* = 10, 95% CI: 423.58, 467.22) (Figure 3). The corrected QT interval prolonged by 20.79 ms from day 1 to day 5 (*p* = 0.13) within this risk group.

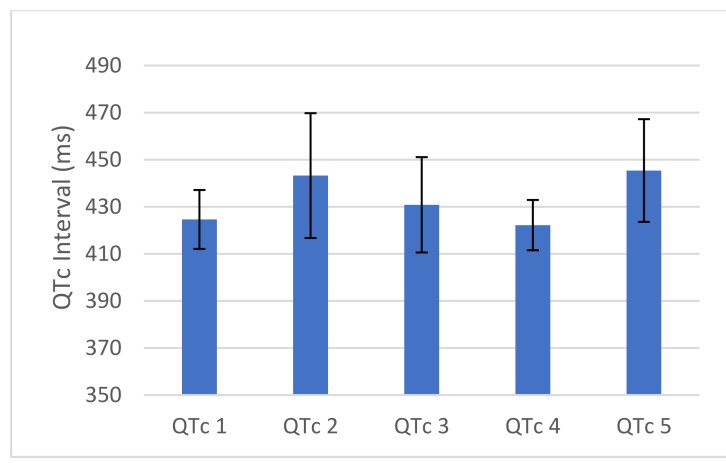

**Figure 3.** Average QTc for patients with Moderate Risk Tisdale Score. QTc reported in ms, *n* = 41, *p* = 0.13 from QTc 1 to QTc 5. Error bars represent 95% Confidence Intervals.

The average corrected QT intervals for patients within the Tisdale Low Risk group were as follows: day 1 of admission was 386 milliseconds (*n* = 8, 95% CI: 368.85, 403.15), day 2 was 372.67 ms (*n* = 3, 95% CI: 370.31, 375.02), day 3 was 410.50 ms (*n* = 4, 95% CI: 357.77, 463.23), day 4 was 398.67 ms (*n* = 3, 95% CI: 393.44, 403.89), and day 5 was 411.50 ms (*n* = 2, 95% CI: 381.12, 396.0) (Figure 4). The corrected QT interval prolonged by 25.50 ms from day 1 to day 5 (*p* = 0.22) within this risk group.

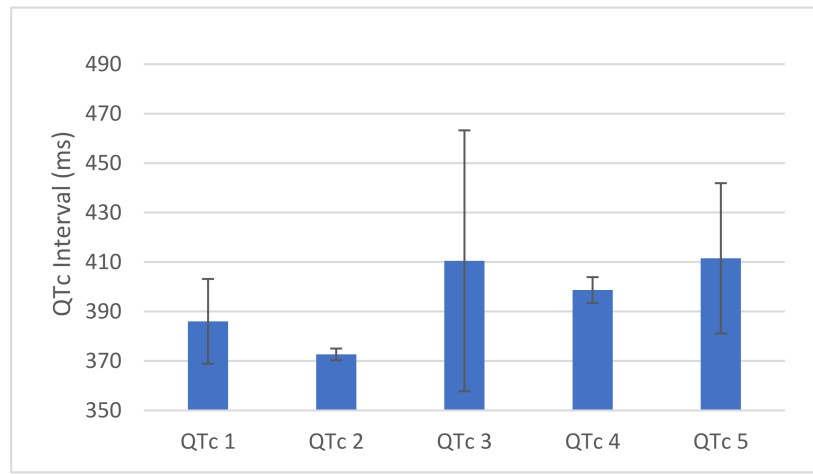

**Figure 4.** Average QTc for patients with Low Risk Tisdale Score. QTc reported in ms, *n* = 9, *p* = 0.22 from QTc 1 to QTc 5. Error bars represent 95% Confidence Intervals.

## 4. Discussion

The use of hydroxychloroquine to treat COVID-19 has been associated with caution and many uncertainties. The use of hydroxychloroquine has also been limited by the potential cardiac side effects, as the medication has the potential to prolong the QT interval. Additionally, patients undergoing the treatment protocol for COVID-19 would also be concomitantly receiving azithromycin, a macrolide antibiotic with its own intrinsic risk of QT interval prolongation.

We sought to determine the cardiac safety of hydroxychloroquine when used as a therapy for COVID-19. In our study, only 1 patient did not complete the hydroxychloroquine treatment protocol due to severe dysphagia. Within the overall population, patients experienced QT prolongation of 20 ms from day 1 to day 5 of treatment. However, we found none of the patients in the study had to discontinue the medication secondary to cardiac arrythmias. Additionally, 72 patients (96%) were concomitantly taking 2 QT prolonging medications and did not experience any TdP. These findings are consistent with previous studies and meta-analyses which have reported an increase in QTc without instances of drug induced TdP [18,19].

The TRS is the only tool that has been developed in a critical care unit and has demonstrated the high-risk category having a sensitivity of 74%, specificity of 77%, positive predictive value of 79%, and negative predictive value of 76% [19]. As we investigated patients within different Tisdale Risk Categories, we found each group experienced similar lengths of QT prolongation, ranging from 21 to 25 ms. There were 66 patients (88%) that were categorized as either moderate or high risk of experiencing QT prolongation according to the Tisdale Risk score. Only 3 of these patients did not receive at least 2 QT prolonging medications. As stated above, there were no patients that were required to discontinue the course of hydroxychloroquine secondary to cardiac arrhythmogenicity or TdP. Interestingly, hypokalemia is associated with increased risk of TdP within the TRS. COVID-19 is known to affect the ACE receptor which has an overall effect within the kidney regarding potassium and sodium balance. Perhaps a correctable risk factor for TdP identified could include COVID-19 patients presenting with hypokalemia.

## 5. Limitations

We found a majority of the patients included in our study did not have their QT intervals documented with daily ECGs within the medical record system. Only 66 patients actually had an ECG reported on day 1 of treatment, and 18 had follow-up ECGs on day 5. Throughout the 5-day course, many physicians would monitor the QT interval by observing the telemetry strip and use that information to guide their decision to order subsequent ECGs. This finding can be explained by the fact that physicians may have decided to utilize this method of QTc monitoring to limit exposure to themselves and other healthcare team members to COVID positive patients. Despite the lack of documented daily ECGs, we did demonstrate the safety of utilizing the 5-day course of hydroxychloroquine in the treatment of COVID-19. Additionally, we only studied a small sample of patients that have been treated within our institution. One meta-analysis performed identified 3 out of 5066 patients who experienced TdP when treated with hydroxychloroquine [20] Including larger number of patients with serial ECGs will be able to provide a more definitive answer to the duration of QT prolongation patients are expected to experience while receiving the 5-day course of hydroxychloroquine. Additionally, our study was limited to the 5-day course of hydroxychloroquine. Subsequent QT monitoring with ECGs after the 5-day course would have provided long term insight into hydroxychloroquine's effect on the QTc.

## 6. Conclusions

Overall, we observed an increase in QTc intervals after patients received the 5-day course of hydroxychloroquine for treatment of COVID-19 without deleterious cardiac side effects. We utilized the Tisdale Risk Score to risk stratify the patients who underwent treatment and patients in each category (low, moderate, or high risk) demonstrated a similar extent of QTc prolongation. Regardless of calculated Tisdale Risk category, there were no patients who had to terminate their treatment with hydroxychloroquine due to arrhythmogenicity or instances of TdP. Further studies of the efficacy and safety of these medication regimens are needed as we continue to care for patients with COVID-19.

**Author Contributions:** Conceptualization, M.T.Z., R.S. and M.L.; Data curation, M.T.Z. and M.L.; Formal analysis, M.T.Z.; Investigation, M.T.Z.; Methodology, M.T.Z.; Project administration, M.T.Z.; Supervision, M.E.Z.; Writing—original draft, M.T.Z. and R.S.; Writing—review & editing, M.T.Z., R.S., M.L. and M.E.Z. All authors have read and agreed to the published version of the manuscript.

**Funding:** This research received no external funding.

**Institutional Review Board Statement:** IRB Study # 1604942-1. Protocol was approved EXEMPT by the chair of the IRB on 5/11/20. Study fulfills requirements for Ascension Providence IRB.

**Informed Consent Statement:** Informed consent not required as this was retrospective study.

**Data Availability Statement:** Data available upon request.

**Conflicts of Interest:** The authors declare no conflict of interest.

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
