# Peer review of "Impact of Hydroxychloroquine Treatment of COVID-19 on Cardiac Conduction: The Beat Goes On"

_covid, doi:10.3390/covid1020039_

Round 1
Reviewer 1 Report
CQ and HCQ have been found to be unable to block the SARS-COV-2 infection in human lung cells, showing that CQ and HCQ have no ability to prevent SARS-CoV-2 from affecting the lungs of severe COVID-19 patients. Previous study was suggested that CQ and HCQ are not beneficial antiviral drugs for curing patients with severe COVID-19. The treatment effect of CQ and HCQ is not only null but also causes serious side effects, which may cause potential cardiotoxicity in severe COVID-19 patients.
This is a very interesting research and well conducted. I believe the manuscript is worth to be published. I have not major comments or suggestions for Authors. Those references should be cited in the manuscript.
Hoffmann, M.; Mösbauer, K.; Hofmann-Winkler, H.; Kaul, A.; Kleine-Weber, H.; Krüger, N.; Gassen, N.C.; Müller, M.A.; Drosten, C.; Pöhlmann, S. Chloroquine does not inhibit infection of human lung cells with SARS-CoV-2. Nat. Cell Biol. 2020, 585, 1–5.
Ho TC, Wang YH, Chen YL, et al. Chloroquine and Hydroxychloroquine: Efficacy in the Treatment of the COVID-19. Pathogens. 2021;10(2):217. Published 2021 Feb 17. doi:10.3390/pathogens10020217
Author Response
Response to Reviewer 1:
Thank you for kindly taking the time to review our manuscript. Your input is valuable and well appreciated. We have included the references into our manuscript within the section regarding questionable to null benefit of HCQ and CQ throughout the literature.
Best,
Marc Zughaib, DO
Reviewer 2 Report
Cardiac toxicity of hydroxychloroquine associated or not with azithromycin was reported and discussed several times in the literature. Prolonged QTc interval is the most frequent ECG abnormality recorded in COVID patient treated with HCQ/AZT but there are many reports and meta-analysis reporting that death related to torsade de pointe is not significantly different from the baseline [1]. Prevalence of torsade de pointe (TdP) is a very rare phenomenon 1/3.000 to 1/10.000 in the USA [2]. A recent paper from Sweden evaluates the prevalence of torsade de point in patient taking drugs that prolonged QT . Torsade de pointe reported with hydroxychloroquine as a prevalence of 21/100,000 treated patient while that of Moxifloxacin, a frequently used quinolone, is 15/ 100,000 in elderly up to 65 year-old [3]. This indicate that the chance to detect a TdP in a patient treated with hydroxychloroquine need to investigate 10,000 patients.
The most interesting in this paper, that merits publication, is the use of the screening score of Tisdale. In this score one of the associated factors which may be corrected , is serum potassium level lower than 3.5 mmol/l. In this study 11 patients had a low serum potassium level. This is a specificity of COVID infection that led to hyper aldosteronism because ACE2 is a main component of the renin-angiotensin system which maintains fluid and salt balance, as well as blood pressure homeostasis. By correcting the potassium level, you reduce the risk of prolonged QTc and that of TdP.
The reviewer thing that it would interesting to compare the mean QTc of different day between the group low medium and high risk to see if this is significantly different which would emphasize on the fact that using this score would be helpful before the introduction of HCQ but also any of the drugs that prolong the QTc including quinolones antibiotics. The mean QTc in the high risk is over 400 ms while in the low risk it is mostly under 400 ms.
Specific comments
Line 46-48. Many RCTs (at least 44) have been now released including meta-analysis on evaluation of HCQ +/- AZT. Please update literature.
Line 56: what mean inexact?
Table 2 please add units for K
You can enrich the discussion with commentaries above.
Limits: With the number of patients in this study you would be very unhappy to get one with TdP. Strength: But for a larger use you can recommend using the score and particularly to test potassium levels before prescribing drugs that prolonged the QTc.
Author Response
Thank you for taking the time to review our manuscript and providing several thoughtful comments. Here are our responses to the suggestions provided.
Line 46-48. Many RCTs (at least 44) have been now released including meta-analysis on evaluation of HCQ +/- AZT. Please update literature.
Literature review of meta-analyses were added to the reference list and texts.
Line 56: what mean inexact?
This word was taken out of the manuscript. The sentence reads more scientifically if it says "The relationship between QT prolongation and TdP is complex."
Table 2 please add units for K
Units were added to Table 2. They were mEq/L.
You can enrich the discussion with commentaries above.
These comments were very enriching and thought provoking. The ACE and hypokalemia correctable risk factor was added to the discussion section. Perhaps this is also an area of future research as well.
Limits: With the number of patients in this study you would be very unhappy to get one with TdP. Strength: But for a larger use you can recommend using the score and particularly to test potassium levels before prescribing drugs that prolonged the QTc.
This was added into the limitations section. We added a reference to a meta-analysis of the incidence of TdP within 5000 patients. However, we would like to include the reference associated with your comment "prevalence of 21/100,000 treated patient while that of Moxifloxacin, a frequently used quinolone, is 15/ 100,000 in elderly up to 65 year-old [3]." If you would provide this reference we will be able to add this into our limitations.